# Effects on Serum Inflammatory Cytokines of Cholecalciferol Supplementation in Healthy Subjects with Vitamin D Deficiency

**DOI:** 10.3390/nu14224823

**Published:** 2022-11-14

**Authors:** Angelo Fassio, Davide Gatti, Maurizio Rossini, Davide Bertelle, Riccardo Bixio, Ombretta Viapiana, Stefano Milleri, Camilla Benini, Francesca Pistillo, Giulia Zanetti, Giovanni Adami

**Affiliations:** 1Rheumatology Unit, University of Verona, 37134 Verona, Italy; 2Centro Ricerche Cliniche di Verona, 37134 Verona, Italy

**Keywords:** vitamin D, cholecalciferol, osteoporosis, cytokines, supplementation, immune response, autoimmune diseases

## Abstract

The effects of different cholecalciferol supplementation regimens on serum inflammatory cytokines in healthy subjects with vitamin D deficiency are still lacking. This is a single-center, open-label, randomized, parallel group study involving healthy subjects deficient in vitamin D (baseline 25OHD < 20 ng/mL) receiving oral cholecalciferol with three different dosing regimens: Group A: 10,000 IU/day for 8 weeks followed by 1000 IU/day for 4 weeks; Group B: 50,000 IU/week for 12 weeks and Group C: 100,000 IU every other week for 12 weeks. IL-17A, IL-6, IL-8, IL-10, IL-23 and TNFα were measured at baseline and at week 4, 8, 12, and 16. 75 healthy subjects were enrolled (58.7% female), with an average age of 34.1 ± 10.2 years. No statistical differences were observed among groups at baseline for either IL-6, IL-17A, IL-23, IL-8 or IL-10 at any time point; TNFα was indetectable. Concerning the whole sample, the time trend analysis showed a statistically significant linear trend for decreasing values over the treatment period for IL-6 (*p* = 0.016) and IL-17A (*p* = 0.006), while no significant time trends were observed for the other teste cytokines. No significant differences were found in the serum concentrations of the tested cytokines between week 12 and week 16. In young healthy individuals deficient in vitamin D, cholecalciferol administration showed a decrease in the serum IL-6 and IL-17A concentrations, without marked differences using the three regimens.

## 1. Introduction

Vitamin D is a key factor in the calcium-phosphorous and bone homeostasis [1]. However, it is now becoming clear that the vitamin D metabolome may be involved in several physiological functions, with a pleotropic effect in the human tissues [2]. In particular, the immunomodulatory functions of vitamin D are of growing scientific interest, with increasing clinical and epidemiological data supporting the link between the vitamin D status and the incidence and severity of conditions such as multiple sclerosis, psoriasis, diabetes, rheumatoid arthritis, inflammatory bowel diseases and infectious diseases [3,4,5,6]. In vitro models suggest that vitamin D contributes to shifting the immune status from a proinflammatory state to a tolerogenic one, suppressing the proliferation of T lymphocytes, promoting differentiation of regulatory T cells and modulating cytokine production [7]. Specifically, vitamin D may decrease the expression of interleukin (IL)-2, IL-6, tumor necrosis factor (TNF)-α, IL-17 and IL-21 and induce the expression of IL-4, IL-5, IL-9 and IL-13 cytokines, resulting in a shift from T helper lymphocytes (Th)-1 and Th-17 to Th-2 immune profile [3]. When considering in vivo models and data from clinical studies, the changes in the serum concentrations of inflammatory cytokines after vitamin D supplementation have been studied in different pathological conditions [8,9,10,11], however with inconsistent results. Moreover, data on vitamin D-deficient healthy subjects are currently lacking. Recently, we published data comparing the pharmacokinetic (PK) and pharmacodynamic (PD) profiles of three different cholecalciferol supplementation schedules (daily administration vs. weekly and biweekly boluses, study DIBA/11) [12,13].

In the present study, we report the data on the exploratory outcomes of the study, namely the effects of the three different cholecalciferol supplementation regimens on serum inflammatory cytokines (IL-6, IL-17A, IL-23, IL-8, IL-10 and TNF-α).

## 2. Materials and Methods

### 2.1. Patients and Study Design

This is a single-center, open-label, randomized, parallel group study in male and female healthy subjects that aimed to compare the PK and PD profiles of cholecalciferol (DIBASE^®^, Abiogen Pharma, Italy) administered as repeated once daily (10,000 IU/day for 8 weeks followed by 1000 IU daily for 4 weeks; group A), weekly (50,000 IU/week for 12 weeks; group B) and alternate week doses (100,000 IU every other week for 12 weeks; group C). The regimens adopted in this trial correspond to the highest dosages allowed for cholecalciferol (DIBASE^®^) in Italy, according to its Summary of Product Characteristics (SmPC), for the correction of vitamin D deficiency in adults [14].

The date of the first enrolment visit was 17 October 2017, and the date of the last enrolment visit was 14 February 2018. The date of the last completed visit was 5 June 2018.

This study was approved by the institutional research committee (protocol identification: DIBA/11, EudraCT Number: 2017-000194) and was conducted in accordance with the 1964 Helsinki declaration and its later amendments or comparable ethical standards. Written informed consent was obtained from all individual participants included in the study. The primary objective of this study was to compare the pharmacokinetic profiles of cholecalciferol (DIBASE^®^) and calcium, phosphate and albumin changes when administered as repeated once daily, weekly and alternate week doses in healthy male and female volunteers. The secondary objectives of the study were to assess the PD profiles investigated the changes in vitamin D metabolites, bone turnover markers, parathyroid hormone, fibroblast growth factor-23, and Wnt pathway inhibitors. The full study protocol and the results related to the primary and secondary outcomes have been already reported elsewhere [12,13].

Inclusion criteria were: caucasian females and males, aged between 18 and 60 years old; BMI between 18.5 kg/m^2^ and 28 kg/m^2^; screening 25OHD < 20 ng/mL; negative urine pregnancy test. The main exclusion criteria were: use of any medicinal product (including antibacterial drugs, over-the-counter medication, vitamins and natural products) in the previous 14 days; history of clinically significant gastrointestinal, renal (including renal stone formation), hepatic, pulmonary, endocrine, oncologic, or cardiovascular disease; or history of epilepsy, asthma, diabetes mellitus, psychosis or severe head injury; vitamin D therapy or food supplements applied within two months; metabolic disorders of calcium or bones (including secondary hyperparathyroidism), history of angina pectoris, artificial UVB exposition (solarium) during the previous 14 days.

Subjects were instructed by the Investigator not to change their lifestyle or diet, including exercise, smoking habits, and fiber intake, during the study and to not have sunbaths or use sunlamps. 

### 2.2. Laboratory Analysis

After the screening, eligible patients were randomized in a 1:1:1 ratio. In all groups A, B and C, blood samples were collected pre-dose on each of on baseline and at week 4, 8, 12, and 16.

Patients’ blood was collected in serum-separating tubes (SSTs) with EDTA containing a special gel that separates blood cells from serum. After the serum extraction, with a laboratory centrifuge set at 3000 RPM for 15 min (EBA 200 EBA 200 S-Hettich Zentrifugen, Andreas Hettich GmbH & Co. KG, Tuttlingen, Germany), serum samples are transferred in 2 mL cryo-tubes (CryoGen^®^ Tubes 2D CLEARLine^®^, Biosigma S.p.A Cona, Venice, Italy) and stored at −70 °C until the end of the study, when all of them were assayed for the following serum markers: IL-17A (Human ELISA kit, Diaclone 850.940 version 6, Diaclone SAS Besancon Cedex, France), IL-10 (Human ELISA kit, Diaclone 850.880 version 5, Diaclone SAS Besancon Cedex, France), IL-8 (human ELISA kit, Diaclone 950.050.192 version CE6, Diaclone SAS Besancon Cedex, France), IL-23 (Human ELISA kit, Diaclone 850.920.096 version 7, Diaclone SAS Besancon Cedex, France), IL-6 (Human ELISA kit, Diaclone 950.035 version 9, Diaclone SAS Besancon Cedex, France) and TNFα (Human ELISA kit, Diaclone 950.090 version 9, Diaclone SAS Besancon Cedex, France). Measured by ELISA immunoassays (Diaclone) on the Fully Automated Microplate Analyser Personal LAB (Adaltis Italia, product code 0-2875 Adaltis, S.r.L Milan, Italy).

Samples were processed in a single unit while the machines were set to read the results in duplicates and provide an average of them.

For the laboratory protocols where samples dilution is required the dilution factor was already consider by the assay during the data extraction, so no additional calculations were required.

The overall intra-assay coefficient of variation (CV) and inter-assay coefficient of variation (CV) were, respectively: IL-17A (3.3% and 5.2%), IL-10 (4.3% and 6.3%), IL-8 (4% and 6%), IL-23 (3.4% and 10.1%), IL-6 (4.4% and 9.1%) and TNFα (3.2% and 10.9%). The lower detection limit for the tested cytokines according to the manufacturer’s specifications were, respectively: IL-17A, 2.3 pg/mL; IL-10, 0.98 pg/mL; IL-8, 29 pg/mL; IL-23, 20 pg/mL; IL-6, 0.81 pg/mL; TNFα, 8 pg/mL. Whenever the assays yielded a value lower than the detection limit, a value equal to the lower detection limit was adopted for statistical purposes.

### 2.3. Statistical Analysis

Analysis of variance (ANOVA) with post hoc analysis (Bonferroni) was used to estimate the absolute differences among groups at baseline.

To test the hypothesis of equal means of cytokines concentrations over the different observation points, one-way ANOVA for repeated measures was adopted, including the treatment period (baseline, week 4, 8 and 12) and the subsequent follow-up to week 16. Sphericity was tested through Mauchly’s test, in the case of violation of the assumption of sphericity, the Huynh-Feldt adjustment was adopted; subsequent pairwise comparisons at different time points were assessed with Bonferroni method for multiple comparisons. Trend analysis was also performed to test for the presence of a linear or quadratic pattern. A two-way ANOVA for repeated measures was used to investigate the interaction of the different supplementation regimens (group A, B and C) over time.

Two-sided *p* values of 0.05 or lower were considered significant. Data are presented as mean ± SD. The sample size of 25 subjects per arm (total N = 75) was set for the primary endpoint of the study (25OHD exposure) and based on clinical judgement and practical considerations and not on formal statistical reasoning.

## 3. Results

In total, 251 healthy volunteers were screened for eligibility. Of these, 75 subjects were randomized to treatment (25 in each treatment arm). 73 volunteers completed the study and 2 prematurely discontinued (Appendix A). One subject in Treatment A discontinued due to adverse event (reported as skin rash with mild severity and not related to the treatment), and 1 subject in Treatment C discontinued due to withdrawal of consent. Compliance to the treatment was 100% for each group at each evaluation. Baseline demographic characteristics and the cytokines serum concentrations at all the observation points are reported in Table 1 and Table 2, respectively.

No statistical differences were observed among treatment groups at baseline for demographic characteristics or for IL-6, IL-17A, IL-23, IL-8, IL-10. For the whole sample, the one-way repeated measures ANOVA showed a statistically result for IL-6 (F = 6.02, *p* = 0.002) and for IL-17A (F = 4.50, *p* = 0.007).

Pairwise comparisons versus baseline showed significant decreases at week 8 and week 16 for IL-6 (*p* = 0.005 and *p* = 0.008, respectively) and at week 4 for IL-17A (*p* = 0.040). Temporal trend analysis indicated a statistically significant linear decreasing pattern for IL-6 (F = 13.43, partial η^2^ = 0.16, *p* = 0.001) and a statistically significant quadratic pattern for IL-17A (F = 8.63, partial η^2^ = 0.14, *p* = 0.005) (Figure 1).

No significant differences were found for cytokines levels among the three groups at any time point of evaluation (time * treatment interaction *p* = 0.427).

TNF-α was undetectable for each patient in each observation (the test was repeated a second time to exclude analytic issues, with the same result). 64 assays for IL-17A (6 assays at baseline, 12 at week 4, 15 at week 8, 16 at week 12 and 15 at week 16), 172 assays for IL-6 (26 assays at baseline, 29 at week 4, 39 at week 8, 42 at week 12 and 36 at week 16), 104 assays for IL-23 (21 assays at baseline, 22 at week 4, 21 at week 8, 20 at week 12 and 20 at week 16), 223 assays for IL-8 (46 assays at baseline, 45 at week 4, 44 at week 8, 44 at week 12 and 44 at week 16) and 233 assays for IL-10 (45 assays at baseline, 45 at week 4, 51 at week 8, 45 at week 12 and 47 at week 16) yielded a result lower than the detection limit and where therefore replaced with the lower detection limit value.

No significant differences were found in the serum concentrations of the tested cytokines between week 12 and week 16 (follow-up period without treatment).

## 4. Discussion

In this study we investigated the effects of cholecalciferol supplementation on serum inflammatory cytokines in a young and healthy vitamin D-deficient population and compared the effects of three different administration schedules. To the best of our knowledge this is the first study with this objective. We observed a decrease in serum IL-6 and IL-17 over time, without differences among the three different administration schedules. Furthermore, no changes in IL-23, IL-8 and IL-10 were documented.

Differences in the PK profile among the three different groups have already been reported in our previous paper [12], which showed a greater systemic 25OHD exposure in the group treated with the daily dose, albeit a quick normalization 25OHD levels was reached in each group. Conversely, we observed a significant effect on PD parameters in the overall cohort, but we did not find significant differences among the three treatment arms on vitamin D metabolites, bone turnover markers, PTH, FGF-23 and Wnt pathway inhibitors [13].

Overall, epidemiological and basic studies have repeatedly suggested a significant association between a poor vitamin status and many immune-related diseases, such as asthma, atherosclerosis, type 2 diabetes, autoimmune diseases [15], and unfavourable clinical outcomes in COVID19, a condition characterised by a dysregulated immune response in its most severe forms [6], but also with other chronic diseases characterized by a proinflammatory status including cardiovascular disease [16], chronic periodontitis [17], chronic plaque psoriasis [18] and metabolic syndrome [19]. In addition, very recently, the data from the VITAL randomised controlled trial showed a 22% reduction in the incidence of autoimmune diseases in subjects supplemented with cholecalciferol (OR 0.78; 95%CI 0.61–0.99) [20].

Moreover, data from RCTs somewhat suggest a potential benefit of cholecalciferol supplementation in different diseases featuring a dysregulated immune system, such as cancer [21], resolution of the inflammatory responses during tuberculosis treatment [5], and respiratory infections [4]. Furthermore, a poor vitamin status has been proposed both as a marker of disease activity in inflammatory bowel diseases and as a predictor of poor response to biological therapy [22], supporting its role as an immunomodulatory hormone with an anti-inflammatory effect, mediated through a reduction of proinflammatory cytokines (IL-2, IL-6, TNF-α and IL-17) and an upregulation of cytokines such as IL-4, IL-5 and IL-13 [3].

Concerning rheumatic diseases, similarly, a low level of serum 25OHD has been associated with the presence, severity and disease activity of rheumatoid arthritis (RA) [23], systemic lupus erythematosus, spondyloarthropathies (SpA) [24] and osteoarthritis, although the benefits of vitamin D supplement for the treatment and prevention of these diseases are still unclear [25]. However, the decrease of serum in IL-6 and IL-17 observed in the present study might support a possible role of vitamin D supplementation in patients suffering from rheumatic diseases, in which IL-6 [26] and IL-17 [27] are key players in their pathogenesis. It could be possible to speculate that correction of the insufficient vitamin D status could optimize the therapeutic response to disease modifying rheumatic drugs. In patients with rheumatoid arthritis, for example, Adami et al. demonstrate that vitamin D supplementation appears to have different effects on pain and disease activity depending on 25OHD serum levels [28], almost proposing vitamin D as a combination therapy to conventional and biologic drugs, even if the current overall evidence is still not strong enough [29].

Our results are also in line with preclinical studies demonstrating a downregulation of proinflammatory cytokines production following treatment with vitamin D [30,31,32]. Corrado et al. recently demonstrated that 25OHD supplementation was associated with a significant reduction of IL-17A and pro-fibrotic cytokines (FGF2, TGFβ, CTGF) both in patients with systemic sclerosis and in healthy subjects, with a dose-dependent effect [33]. On the other hand, clinical data showed inconsistent results on serum IL-17 and Th-17 cells in subjects with neurologic conditions [8,9,10]. As possible explanation for these inconsistent results, however, is the bias linked to the choice of immune-assays investigations, which can be a relevant confounder for the reproducibility of individual study, as demonstrated in patients with multiple sclerosis [34].

A systematic review published in 2017 that included 81 trials did not confirm significant effects of vitamin D supplementation on inflammatory biomarkers, including C-reactive protein, IL-6 and TNFα, in several pathological conditions (but no study on healthy subjects was included) [11]. Concerning IL-6 only one trial [35] suggested that supplementation of 25OHD decreased significantly its serum concentration in obese subjects, albeit the overall evidence does not corroborate a significant impact of vitamin D supplementation on this biomarker in the obese [36].

On the other hand, the Biochemical Efficacy and Safety Trial of Vitamin D (BEST-D), a double-blind, placebo-controlled, dose-finding, randomized clinical trial, showed no significant change in gene expression and serum levels of IL-6, IL-8, IL-10, TNFα and IFN-γ in healthy subject receiving supplementation with 25OHD or placebo for 12 months. BEST-D’s population, however, was older (mean age 72 years) than our cohort, with slightly higher baseline vitamin D levels (mean 20 ng/mL) and longer serological control. IL-17 was not dosed. Our study, otherwise, focused on healthy young adults; also our results of a significant reduction of IL-6 and IL-17A are concentrated in the first weeks of treatment [37].

Motamed et al. published results in line with those of the present study on pregnant women, showing that supplementation with 2000 IU/d vitamin D3 is more effective than 1000 IU/d in terms of increasing circulating 25OHD, ameliorating inflammatory markers as TNFα and IL-6, and improving neonatal outcomes, although the basal levels of vitamin D were not uniform [38].

In general, although a correlation between low vitamin D levels and various pathological conditions and the reduction of all-cause mortality in older people with its supplementation have been observed [39], the benefits of systematic cholecalciferol supplementation on non-skeletal disorders remain controversial, as low 25OHD might also be the consequence of ill health, rather than its cause [11].

Our study does have some limitations that need to be mentioned. First, though the trial design included three groups randomised to different cholecalciferol regimens, it lacks a placebo group as a reference. Second, we measured serum cytokine levels immunoenzymatic reaction (ELISA), a method that can be affected by different variables (i.e., pregnancy, severe illnesses) [40,41] and dose biomarkers that are currently unstandardised. However, we studied a healthy (although vitamin D deficient) population and for this reason we do not expect these issues to undermine our results.

## 5. Conclusions

In conclusion, in our cohort of young, healthy vitamin D deficient subjects, we observed a decrease in the serum concentrations of IL-6 and IL-17A, without marked differences among the three different regimens of supplementation.

## Figures and Tables

**Figure 1 nutrients-14-04823-f001:**
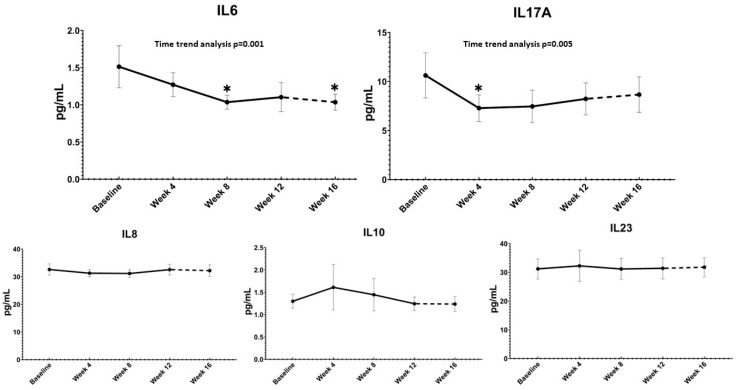
Absolute changes with time trend analysis of the overall cohort over time of IL-6, IL-17A, IL-10, IL-8 and IL-23. Treatment was administered from baseline to week 12 (solid line), the follow-up period without supplementation (from week 12 to week 16) is depicted by the dashed line. * *p* < 0.05 with respect to baseline (one-way ANOVA for repeated measures). Error bars show 95% Confidence Interval.

**Table 1 nutrients-14-04823-t001:** Subjects demographic at baseline. No statistical differences were observed among groups. BMI = body mass index; SD = standard deviation.

Baseline Characteristics	All Subjects (*n* = 75)	Group A (*n* = 25)	Group B (*n* = 25)	Group C (*n* = 25)
Sex				
Male; N (%)	31 (41.3%)	12 (48%)	7 (28%)	12 (48%)
Female; N (%)	44 (58.7%)	13 (52%)	18 (72%)	13 (52%)
Age (years); Years (SD)	34.1 (10.2)	30.2 (10)	36.7 (8.8)	35.4 (11)
Weight (Kg); Mean (SD)	66.7 (12.4)	65.2 (13.5)	67.4 (9.8)	67.6 (13.7)
BMI (Kg/m^2^); Mean (SD)	23.1 (2.6)	22.6 (2.9)	23.4 (2.1)	23.2 (2.8)

**Table 2 nutrients-14-04823-t002:** Serum cytokines concentrations at baseline. No statistical differences were observed among groups.

	All Subjects (*n* = 73)	Group A (*n* = 24)	Group B (*n* = 25)	Group C (*n* = 24)
**IL-6 (pg/mL)**				
**Baseline**				
N	73	24	25	24
Median (IQR)	0.99 (0.81–2.43)	1.12 (0.81–1.57)	2.33 (1.04–5.85)	0.84 (0.81–3.02)
Min-Max	0.81–6.81	0.81–2.38	0.82–6.81	0.81–4.64
**Week 4**				
N	73	24	25	24
Median (IQR)	0.92 (0. 81–1.84)	1.40 (0.81–1.70)	1.01 (0.81–1.64)	1.19 (0.81–2.56)
Min-Max	0.81–3.72	0.81–3.72	0.81–3.27	0.81–3.11
**Week 8**				
N	73	24	25	24
Median (IQR)	0.81 (0. 81–1.53)	0.81 (0.81–1.17)	0.89 (0.81–1.66)	0.9 (0.81–2.28)
Min-Max	0.81–2.55	0.81–2.55	0.81–1.89	0.81–2.51
**Week 12**				
N	73	24	25	24
Median (IQR)	0.81 (0. 81–1.18)	0.81 (0.81–1.10)	0.86 (0.81–1.57)	0.95 (0.81–1.37)
Min-Max	0.81–1.79	0.81–1.54	0.81–1.79	0.81–1.74
**IL-17A (pg/mL)**				
**Baseline**				
N	73	24	25	24
Median (IQR)	5.57 (4.45–13.38)	5.54 (4.17–8.52)	5.61 (3.87–10.60)	9.92 (4.28–16.83)
Min-Max	2.67–20.45	2.67–20.45	3.3–12.25	2.91–20.36
**Week 4**				
N	73	24	25	24
Median (IQR)	4.05 (2.32–11.43)	3.71 (2.46–12.91)	3.81 (2.3–8.79)	7.03 (2.34–15.16)
Min-Max	2.3–18.40	2.3–17.88	2.3–9.95	2.3–18.40
**Week 8**				
N	73	24	25	24
Median (IQR)	3.85 (2.30–10.48)	3.85 (2.3–11.87)	3.43 (2.3–4.63)	6.18 (2.93–18.52)
Min-Max	2.3–21.01	2.3–16.37	2.3–5.65	2.3–21.01
**Week 12**				
N	73	24	25	24
Median (IQR)	5.54 (2.50–13.66)	5.88 (2.69–11.87)	3.52 (2.3–15.17)	9.41 (2.59–18.09)
Min-Max	2.3–18.76	2.3–15.44	2.3–18.64	2.3–18.76
**IL-23 (pg/mL)**				
**Baseline**				
N	73	24	25	24
Median (IQR)	37.76 (20–45.45)	37.76 (20–46.54)	45.91 (24.62–57.36)	34.11 (20–38.86)
Min-Max	20–60.4	20–48.7	20–60.4	20–52.32
**Week 4**				
N	73	24	25	24
Median (IQR)	35.57 (20–41.28)	35.57 (20–42.15)	37.76 (23.71–36.48)	35.57 (20–40.69)
Min-Max	20–129.65	20–129.65	20–37.03	20–45.08
**Week 8**				
N	73	24	25	24
Median (IQR)	36.30 (20–42.63)	36.3 (20–37.03)	39.33 (23.89–53.67)	38.13 (20–50.39)
Min-Max	20–56.77	20–37.76	20–56.77	20–55.35
**Week 12**				
N	73	24	25	24
Median (IQR)	36.29 (20–40.32)	35.66 (20–39.95)	39.07 (24.55–39.77)	34.11 (20–48.74)
Min-Max	20–55.35	20–40.68	20–39.95	20–55.35
**IL-8 (pg/mL)**				
**Baseline**				
N	73	24	25	24
Median (IQR)	29 (29–34.27)	29 (29–32.19)	32.67 (29–52.29)	29 (29–32.73)
Min-Max	29–63.35	29–62.43	29–63.35	29–43.91
**Week 4**				
N	73	24	25	24
Median (IQR)	29 (29-39.17)	35.33 (29–49.96)	29 (29–39.86)	29 (29–34.90)
Min-Max	29–51.61	29–51.61	29–43.91	29–43.01
**Week 8**				
N	73	24	25	24
Median (IQR)	29 (29–35.34)	29 (29–33.25)	29 (29–32.98)	33.76 (29–41.22)
Min-Max	29–49.96	29–36.34	29–38.31	29–49.96
**Week 12**				
N	73	24	25	24
Median (IQR)	29 (29–41.69)	29 (29–29)	33.34 (29–42.58)	39.01 (29–55.61)
Min-Max	29–57.93	29–24.02	29–45.69	29–57.93
**IL-10 (pmol/L)**				
**Baseline**				
N	73	24	25	24
Median (IQR)	1.52 (0.98–1.81)	1.27 (0.98–1.95)	1.94 (1.13–2.44)	1.25 (0.98–1.59)
Min-Max	0.98–2.78	0.98–2.78	0.98–2.68	0.98–2.62
**Week 4**				
N	73	24	25	24
Median (IQR)	0.98 (0.98–1.82)	0.98 (0.98–1.89)	1.18 (0.98–2.48)	1.31 (0.98–2.06)
Min-Max	0.98–2.73	0.98–2.41	0.98–2.73	0.98–2.46
**Week 8**				
N	73	24	25	24
Median (IQR)	1.02 (0.98–2.28)	1.02 (0.98–1.36)	1.31 (0.98–4.66)	1.56 (0.98–2.47)
Min-Max	0.98–6.34	0.98–6.34	0.98–5.67	0.98–2.58
**Week 12**				
N	73	24	25	24
Median (IQR)	0.98 (0.98–1.15)	1.01 (0.98–2.32)	0.98 (0.98–3.00)	0.98 (0.98–1.24)
Min-Max	0.98–3.86	0.98–3.86	0.98–3.64	0.98–1.29

Comparisons were performed between groups (A vs. B vs. C) by analysis of variance; IL = interleukin; IQR = interquartile range; SD = standard deviation.

## Data Availability

Data can be made available from the corresponding author upon request.

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
