# Peer review of "Effects on Serum Inflammatory Cytokines of Cholecalciferol Supplementation in Healthy Subjects with Vitamin D Deficiency"

_nutrients, 2022, doi:10.3390/nu14224823_

Round 1
Reviewer 1 Report
The study of Fassio et al is clearly designed and well written. There is much interest in the potential effects of vitD on the immune system and in this study the effect is analysed with respect to inflammatory cytokines in the circulation. In general, cytokine levels in the circulation are rather low and, hence, close to the detection limits of the available assays. As such, changes need to be interpreted with caution.
Major comment:
Since OD-values of ELISA do not have a linear relation with concentrations, the reliability of results in the lower range of detection becomes tricky. Extrapolation of results below the lowest standard (not the 0-standard) has its limitations, but according to the inserts of the kits the lower detection limit is somewhat below this lowest standard. However, in the current study levels below this lower detection limit are analysed. These data should not be included as they are. For instance, the detection limit of IL-17 is 2.3 pg/mL, while the minimum value is reported as 0.16 pg/mL; similarly for IL-6 with a limit of 0.81 pg/mL and a minimum reported value of 0.03 pg/mL. Overall, this implies that the statement in the results that in all samples these cytokines could be detected (lines 150-151) is not correct and that the results should be analysed accordingly.
Minor comments:
1. Please provide additional information about the cytokine assays: measured in serum? samples in endotoxin free tubes? duplicates? dilution? If diluted, is dilutionfactor used in the calculation of the final concentration? what is the detection range?
2. Typing error in table 1: Wight should be Weight.
3. Table 2 presents data both as mean (SD) as well as median (IQR). If data are normally distributed mean (SD) should be used; if data are not normaly distributed (or not checked for normality) median (range) should be used.
4. Although data on 25(OH)D are presented in a previous paper, it might be considered to include a figure with these data in order to have these relevant data directly available when reading this paper.
5. In the discussion (lines 211-212) there is a remark on inconsistency in results of IL-17. In this context the paper of Rolf et al (Mol Immunol 2019; 105: 198-204 would be interesting to add.
6. In the discussion (line 216) there is a remark that data from healthy subjects are missing in the systematic review. In this context the paper of Berlanga-Taylor et al (eBioMedicine 2018;31:133-142) will be highly valuable to discuss.
7. Typing error in line 239: IL-L instead of IL-6.
Reviewer 2 Report
This article reveals that supplements of cholecalciferol to healthy people with vitamin D deficiency can reduce the content of IL-17A and IL-6 in the blood. Grammar and typing errors should be revised throughout this manuscript, especially in these tables.
1. In the manuscript, there are wrong rows in the table. For example, the row in the table is not aligned. In table 2, N corresponds to 73, 24, 25, and 24.
2. In table 2, pg/ml should be changed to pg/mL.
3. In line 81, kg/m2 should be written as kg/m2.
4. Figure 1 shows the detection of inflammatory factors at different time points. In the manuscript, we do not see which group of results is detected in Figure 1.
5. IL-17A was detected in this article, and IL-17 written in lines 27, and 239 should be changed to IL-17A.
6. There are many inconsistencies in writing inflammatory factors in manuscripts, such as IL-6 and IL6, IL-8, and IL8.
7. In the conclusions section, check the spelling errors of IL-L.
Author Response
We thank the Reviewers for their positive comments and their helpful suggestions and advice. We implemented all the changes suggested and we addressed each point raised.
Reviewer: 2
Comments: This article reveals that supplements of cholecalciferol to healthy people with vitamin D deficiency can reduce the content of IL-17A and IL-6 in the blood. Grammar and typing errors should be revised throughout this manuscript, especially in these tables.
We thank the Referee for the comment. We revised typing errors and adjusted errors in the tables.
1. In the manuscript, there are wrong rows in the table. For example, the row in the table is not aligned. In table 2, N corresponds to 73, 24, 25, and 24.
Amended
2. In table 2, pg/ml should be changed to pg/mL.
Amended
3. In line 81, kg/m2 should be written as kg/m².
Amended
4. Figure 1 shows the detection of inflammatory factors at different time points. In the manuscript, we do not see which group of results is detected in Figure 1.
We described results considered in Figure 1 in the results section, with the sentence “Pairwise comparisons versus baseline showed significant decreases at week 8 and week 16 for IL-6 (p=0.005 and p=0.008, respectively) and at week 4 for IL-17A (p = 0.040). Temporal trend analysis indicated a statistically significant linear decreasing pattern for IL-6 (F=13.43, partial η2=0.16, p=0.001) and a statistically significant quadratic pattern for IL-17A (F=8.63, partial η2=0.14, p=0.005) (Figure 1)”.
5. IL-17A was detected in this article, and IL-17 written in lines 27, and 239 should be changed to IL-17A.
Amended
6. There are many inconsistencies in writing inflammatory factors in manuscripts, such as IL-6 and IL6, IL-8, and IL8.
Amended
7. In the conclusions section, check the spelling errors of IL-L.
Amended
Round 2
Reviewer 1 Report
Thank you for the adjustments. I do not have any further comments.
Reviewer 2 Report
This reviewer accepts the revised version